# Non-Targeted Metabolomics of Serum Reveals Biomarkers Associated with Body Weight in Wumeng Black-Bone Chickens

**DOI:** 10.3390/ani14182743

**Published:** 2024-09-23

**Authors:** Zhong Wang, Xuan Yu, Shenghong Yang, Mingming Zhao, Liqi Wang

**Affiliations:** 1Key Laboratory of Animal Genetics, Breeding and Reproduction in the Plateau Mountainous Region, Ministry of Education, Guizhou University, Guiyang 550025, China; zwang8@gzu.edu.cn (Z.W.); 15285059211@139.com (X.Y.); yangshenghong2023@163.com (S.Y.); 18385046180@163.com (M.Z.); 2College of Animal Sciences, Guizhou University, Guiyang 550025, China

**Keywords:** Wumeng black-bone chicken, body weight, metabolomics, riboflavin, 2-isopropylmalic acid, inflammation

## Abstract

**Simple Summary:**

Growth traits are important economic traits of broilers. We found the key metabolites that may be related to the body weight of Wumeng black-bone chicken by comparing their serum metabolomics between the high- and low-weight groups. Metabolites with high levels in serum such as riboflavin and 2-isopropylmalic acid may be beneficial to growth performance, and the enrichment of inflammation-related metabolites may be related to low body weight of chickens. Our results can provide a certain reference for the analysis of metabolic mechanism of body-weight formation of chickens.

**Abstract:**

Growth performance is an important economic trait of broilers but the related serum metabolomics remains unclear. In this study, we utilized non-targeted metabolomics using ultra-high-performance liquid phase tandem mass spectrometry (UHPLC-MS/MS) to establish metabolite profiling in the serum of Chinese Wumeng black-bone chickens. The biomarker metabolites in serum associated with growth performance of chickens were identified by comparing the serum metabolome differences between chickens that significantly differed in their weights at 160 days of age when fed identical diets. A total of 766 metabolites were identified including 13 differential metabolite classes such as lipids and lipid-like molecules, organic acids and their derivatives, and organoheterocyclic compounds. The results of difference analysis using a partial least squares discriminant analysis (PLS-DA) model indicated that the low-body-weight group could be differentiated based on inflammatory markers including prostaglandin a2, kynurenic acid and fatty acid esters of hydroxy fatty acids (FAHFA), and inflammation-related metabolic pathways including tryptophan and arachidonic acid metabolism. In contrast, the sera of high-body-weight chickens were enriched for riboflavin and 2-isopropylmalic acid and for metabolic pathways including riboflavin metabolism, acetyl group transfer into mitochondria, and the tricarboxylic acid (TCA) cycle. These results provide new insights into the practical application of improving the growth performance of local chickens.

## 1. Introduction

As the global population expands, chickens will undoubtably be served as a primary source of high-quality animal protein [1]. Growth indicators in chickens are also linked to important economic traits [2] and genetic improvements of growth traits have successfully relied on traditional breeding methods [3]. Previous research on these traits has focused on the discovery of genetic markers[4] associated with growth traits [5,6,7]. These markers include *ACTA1* encoding skeletal α-actin [8,9], *TGFβ3* encoding transforming growth factor β [10], *MLNR* encoding the gut hormone motilin receptor, and the mediator complex gene named *MED4* that is required for specific transcriptional activation [11]. However, host genome information cannot fully explain the complex phenotype of chicken growth that is also linked to sex and nutritional status as well as disease state [12,13,14]. The advent of metabolomics technologies has provided novel methods to investigate host physiology since metabolites play important intermediate roles between the genome and growth processes [15].

Metabolomics refers to the use of chromatography–mass spectrometry and nuclear magnetic resonance techniques to identify and quantify all small-molecule metabolites (<1000 Da) in biological samples. These metabolites reflect the variations in the animal genome that are transferred to the growth phenotype via transformation of metabolic molecules [16]. The production and metabolism of small molecules revealed by metabolomics can more directly and accurately reflect the animals’ physiological state and phenotype [17]. In particular, circulating blood conveys metabolic information and can represent the state of the normal physiology of the host [16]. Metabolomics therefore can provide new insights for the analysis of animal traits and may serve to identify small-molecule metabolites that are closely linked to biomarkers related to meat quality [18], feed intake [19], growth performance [20], and disease [21]. A previous study on growth performance of chickens revealed that residual feed intake and serum uric acid were significantly positively correlated [22]. The combination of gut microbiome and metabolome analyses also indicated that specific cecal microbiota promoted growth via conjugated linoleic acid (bovinic acid) [20]. High cecal abundance of *Microbacterium* and *Sphingomonas* in chickens also improved growth performance by regulating fat metabolism [23]. However, these studies have not been consistent, so metabolic mechanisms associated with body weight have not been fully revealed.

The Wumeng black-bone chicken is a unique local breed in the southwest of China and possess black coloring of skin, muscle, bone, and internal organs, and is also a consumer favorite [24]. Body weight is a key index of chicken growth performance traits and directly affects the economic benefits of broilers [25]. Therefore, in this study, Wumeng black-bone chickens were taken as the study object to identify key candidate metabolites related to growth performance using serum non-targeted metabolomics. We compared high- and low-body-weight groups using UHPLC-MS/MS. The results of this study will provide a reference for the analysis of chicken growth traits.

## 2. Materials and Methods

### 2.1. Animals and Experimental Design

The chickens were raised in the chicken farm of Guizhou University for scientific research from June 2022 to October 2022. Chickens hatched in the same batch were reared under the same condition. Specifically, the animals were housed in three-layer iron cages with a feeding density of 16 mixed-sex chickens per cage at 0–4 weeks, 8 mixed-sex chickens per cage at 4–10 weeks, and 1 chicken per cage at 10–18 weeks. The chicken house had no temperature control system and a rolling curtain was used for ventilation and temperature control. The daily light duration was about 16 h. During the experimental period, the temperature of the chicken house was 15–35 °C. The chicken houses were cleaned and disinfected regularly according to the sanitary and epidemic prevention requirements of the chicken farm. Chickens were fed with commercial feed in the morning and afternoon every day and were allowed to drink water freely. The ingredient compositions of the diets are shown in Table 1. All chickens were vaccinated according to routine immunization procedures that included Marek’s and Newcastle disease, infectious bronchitis, bursal virus, and avian influenza. A total of 58 chickens were reared to 160 days of age and slaughtered after being weighed, and at this time intravenous blood was collected for serum separation. The serum samples were stored at −80 °C for later analyses. According to the ranking-order of weight, a total of 32 chickens including 16 chickens (half male and half female) with the highest weight and 16 chickens with the lowest weight (half male and half female) were divided into a highbody-weight group (WH) and a low-body-weight group (WL).

### 2.2. Extraction of Serum Samples 

The serum samples were thawed at 4 °C and 100 μL aliquots were added to 400 μL pre-cooled methanol/acetonitrile/H_2_O (*v*/*v*/*v*: 2:2:1), vortex-mixed and ultrasonicated on ice for 30 min, and then left to stand at −20 °C for 10 min. The samples were then centrifuged at 14,000× *g* at 4 °C for 20 min and supernatants were vacuum-dried. The residue was dissolved in 100 μL aliquots of 50% acetonitrile, vortexed, and centrifuged at 14,000× *g* at 4 °C for 15 min and the supernatant was collected for UHPLC-MS/MS analysis.

### 2.3. Ultra-High-Performance Liquid Chromatography–Mass Spectrometry Analysis

#### 2.3.1. Chromatographic Conditions

The samples were separated on an Agilent 1290 Infinity ultra-high-performance liquid chromatography system using an HILIC column at 25 °C with a flow rate of 0.5 mL/min and injection volume of 2 μL. Water containing 25 mM ammonium acetate and 25 mM ammonia was taken as mobile phase A and acetonitrile was taken as mobile phase B. Analytes were separated by gradient elution as follows: 0–0.5 min, 95% mobile phase B; 0.5–7.0 min, 95–65% mobile phase B; 7.0–8.0 min, 65–40% mobile phase B; 8.0–9.0 min, 40% mobile phase B; 9–9.1 min, 40–95% mobile phase B; 9.1–12 min, 95% mobile phase B. The temperature of the auto-sampler was set to 4 °C. Quality control samples were inserted into the sample queue to guarantee data reliability.

#### 2.3.2. Quadrupole Time-of-Flight Mass Spectrometry (Q-TOF-MS) Conditions

Following UPHLC separation, the samples were analyzed by mass spectrometry using a Q-EX Active series mass spectrometer (Thermo, Waltham, MA, USA). The positive and negative ion modes of electrospray ionization were employed. The ESI source and mass spectrometry parameters were set as follows: Gas 1, Gas 2, and air curtain gas were 60, 60, and 30 psi, respectively; ion source temperature and spray voltage were 600 °C and ±5500 V (positive and negative modes), respectively. In MS only acquisition, *m*/*z* range, resolution, and scanning accumulation time were set at 80–1200 Da, 60,000, and 100 ms, respectively. In auto MS/MS acquisition, *m*/*z* range, resolution, and scanning accumulation time and dynamic exclusion time were set at 70–1200 Da, 30,000, and 50 ms and 4 s, respectively.

#### 2.3.3. Analysis of Metabolome Data 

Raw MS data were converted into MzXML files with ProteoWizard MSConvert before being imported into XCMS software (Version 3.7.1)[26]. Peak alignment, retention time correction, and peak area extraction were conducted. centWave *m*/*z* = 10 ppm, peakwidth = c (10, 60), prefilter = c (10, 100). Parameters for peak pick were centWave *m*/*z* = 10 ppm, peakwidth = c (10, 60), and prefilter = c (10, 100). bw = 5, mzwid = 0.025, and minfrac = 0.5 were used for peak clustering. An internal MS2 database (BiotreeDB) was used to annotate metabolites and the HMDB database was used to identify metabolites. Multivariate analysis of the final data of peak number, sample name and normalized peak area was performed using SIMCA 16.0.2 (Sartorius Stedim Data Analytics AB, Umea, Sweden) [27].

### 2.4. Statistical Analysis

The online analysis platform MetaboAnalyst 5.0 [28] (https://www.metaboanalyst.ca/) (accessed on 15 January 2024) was employed to analyze serum metabolomics using partial least squares discriminant analysis, Student’s t test (T-test), and metabolite enrichment analysis. The Wilcoxon test was used to compare the body weight of chickens in the WH and WL groups. 

## 3. Results 

### 3.1. Body Weights of Chickens at 160 Days of Age 

The average body weight of Wumeng black-bone chickens at 160 days of age was 1675 ± 308 g (*n* = 58). The Shapiro–Wilk normal distribution test indicated that the body weights conformed to a normal distribution (W = 0.97, *p* = 0.1, Appendix A). The average body weight of the WH group was 1790 ± 393 g (*n* = 16) and significantly (*p* < 0.05) higher than that of the WL group (1459 ± 272 g, *n* = 16) (Figure 1A,B and Appendix A).

### 3.2. Serum Metabolome Profile of Wumeng Black-Bone Chickens

Non-targeted serum metabolome of the chickens uncovered a total of 11,996 metabolite features including 7086 in the positive and 4910 in the negative ion mode. We were able to identify 766 metabolites by comparison with the metabolomics database; these included 494 positive and 272 negative ion mode metabolites. In particular, we were able to annotate 235 lipids and lipid-like molecules, 131 organic acids and derivatives, 100 organoheterocyclic compounds, 82 benzenoids, 48 organic oxygen compounds, 34 phenylpropanoids and polyketides, 26 organic nitrogen compounds, 16 nucleosides, nucleotides and analogues, 7 alkaloids and derivatives, 2 homogeneous non-metal compounds, 1 organophosphorus compound, and 1 organosulfur compound; 82 metabolites were undefined (Figure 2).

### 3.3. Differential Serum Metabolites between High- and Low-Body-Weight Groups

These metabolomics data were further analyzed using MetaboAnalyst 5.0 and the PLS-DA results indicated that the serum metabolome of chickens in high- and low-body-weight groups were significantly separated in the positive and negative ion modes (Figure 3A,B). Setting the threshold levels of variable importance in the projection (VIP) > 2 and *p* < 0.05 resulted in 30 metabolites identified in the positive ion mode that significantly differed between the WH and WL groups. There were nine metabolites enriched in the serum of chickens in the WH group and riboflavin possessed the highest VIP value (4.6368). In addition, benzamide, 3-pyridinecarboxaldehyde, 1-methyl-2-undecylquinolin-4-one, and taurochenodeoxycholic acid were also enriched in the WH group. Twenty-one metabolites were identified in the sera of the WL group and FAHFA were the most abundant (M632T188, M656T186, M630T187, and M590T190). In addition, the WL group was enriched for the inflammatory mediator prostaglandin a2, the steroid hormone 17-β-nandrolone decanoate, and aflatoxin g1 (Figure 3C and Table 2).

In the negative ion mode, a total of 13 metabolites with significantly different levels in serum between the high- and low-body-weight groups were identified by PLS-DA. There were four metabolites enriched in the WH group that were primarily organic acids. Specifically, 2-isopropylmalic acid possessed the highest VIP value of 5.0179 followed by citrate, 2-(4-chloro-2-methylphenoxy)-acetic acid, and 3, 7-dimethyluric acid. There were nine metabolites enriched in the serum of chickens in the WL group including two bile acids (ursodeoxycholic and glycocholic) and kynurenic acid (a metabolite associated with inflammation), as well as the PAR-2 activating peptide Ser-Leu-Ile-Gly-Lys-Val-amide (Figure 3D and Table 3). 

### 3.4. Enrichment Analysis of Metabolic Pathways of Differential Serum Metabolites

The functional enrichment analysis of the differential metabolites between the WH and WL groups indicated that six metabolic pathways were enriched in the serum of chickens in the WH group, including riboflavin metabolism, transfer of acetyl groups into mitochondria, caffeine metabolism, the TCA cycle, the Warburg effect, and bile acid biosynthesis, while there were five metabolic pathways enriched for the WL group, including inflammatory metabolic pathways (tryptophan and arachidonic acid metabolism) as well as butyrate metabolism and fatty acid and bile acid biosynthesis (Figure 4 and Appendix A).

## 4. Discussion

Growth is a complex economic trait that can be affected by many factors including breed, nutrition, feeding management, and environment. Local chickens are popular with consumers because of their good meat quality and strong resistance to environmental stress. In contrast, compared with highly bred commercial breeds they are slow growers and this restricts industrial production and large-scale promotion of local chickens [29]. Therefore, it is of great practical significance to uncover mechanisms that lead to weight gain for these local chickens. Metabolomics provides a bridge between host genes and phenotype and enables a new perspective for the analysis of genetic mechanisms of chicken growth traits. In this study, we explored potential factors affecting growth and we chose market weight as a research index to establish high- and low-body-weight groups. We subjected sera from these animals to non-targeted metabolomics to identify marker metabolites in chicken serum related to body weight. As a result, the serum metabolome profiles of Wumeng black-bone chickens were described and 13 metabolites were linked to growth improvements. These included riboflavin and 2-isopropylmalic acid, and pathway enrichment analysis uncovered six metabolic pathways correlated with high body weight, including riboflavin metabolism, transfer of acetyl groups into mitochondria, and caffeine metabolism. Inflammation has also been associated with growth inhibition in chickens and metabolites related to inflammation were found in the WL but not the WH group. These metabolites included prostaglandin a2, kynurenic acid, and FAHFA, which correlated with low body weight. We also found five metabolic pathways for the WL group including tryptophan and arachidonic acid metabolism.

Metabolomics differential analysis identified 13 metabolites that might promote the growth of Wumeng black-bone chickens. In the positive ion mode, riboflavin (vitamin B2) showed the highest VIP value for the WH group and riboflavin metabolism was the predominant pathway for these animals, which implicates this vitamin in promoting weight gains. Riboflavin in the form of flavin mononucleotide and flavin adenine dinucleotide participates in electron transfer during redox reactions including fatty acid oxidation, the TCA cycle, mitochondrial respiratory chain electron transfer, and amino acid degradation [30]. Riboflavin supplementation can improve energy metabolism efficiency in mice or humans; the lack of riboflavin in the diet inhibits growth while its addition to diets can significantly improve the growth of poultry [31]. A previous study had evaluated riboflavin supplementation in broiler diets under different nutritional conditions and concluded that the supplemental level of 3.6 mg/kg added in the initial 4 weeks of life was optimal [32]. Therefore, we speculate that riboflavin in serum is a key metabolite affecting chicken body weight and riboflavin supplementation may promote the growth performance of Wumeng black-bone chickens.

In the negative ion mode, 2-isopropylmalic acid was the metabolite with the highest VIP value for the WH group. In mitochondria, malic acid is an intermediate product in the TCA cycle [33] and our enrichment analyses of pathway indicated this pathway was enriched in the WH group. The TCA cycle is the center of cell metabolism that begins with condensation of oxaloacetic acid and acetyl-CoA derived from numerous sites including oxidation of fatty acids, amino acids, or pyruvate to form citric acid. The TCA cycle provides energy and raw materials for survival and growth [33]. Our results suggest that the active metabolism of the TCA cycle is a key factor for chicken growth performance and enhancing its activity thus leads to body-weight increases. Consistent with this finding, a riboflavin deficit leads to mitochondrial energy metabolism dysfunction and obstruction of the TCA cycle that would be detrimental to growth and development [34]. Therefore, we speculate that the TCA cycle is obstructed in low-body-weight chickens due to riboflavin deficiency. Furthermore, there were other organic acids that were enriched in the WH group, including 3,7-dimethyluric acid. The uric acid derivative 3,7-dimethyluric acid was enriched in the WH group and may be indicative of increased protein nitrogen metabolism. The accumulation of these substances in the systemic circulation may therefore contribute to enhancing weight gain but this requires further experimentation. 

Many previous studies have shown that chronic inflammation can severely restrict chicken growth [20,35]. Significantly up-regulated expression of TLR4, NF-κB, MyD88, and other cytokines in the chicken jejunum resulted in impaired intestinal barrier integrity and thus inhibited chicken growth [35]. Prostaglandin a2 is a key lipid mediator of type 2 inflammation and its enrichment in the WL group may indicate the presence of inflammation [36]. The kynurenine pathway is tightly linked to feather pecking behaviors in chickens and may indicate (as in mammals) a stress state promoted by the release of pro-inflammatory cytokines [37]. We also found that FAHFA were enriched in the serum of WL chickens and 4/21 metabolites associated with low body weight in the positive ion mode were FAHFA. In particular, alpha-oxidized products of monocarboxylic fatty acids are a newly discovered class of bioactive lipid molecules with anti-inflammatory effects [38]. Mice fed the fatty acid ester of palmitic and hydroxystearic acids (PAHSA) showed decreased TNF-α and IL-1β expression in macrophages when fed a high-fat diet and PAHSA were linked with obesity-related inflammation [39]. In addition, macrophages stimulated by lipopolysaccharide produced FAHFA molecules with anti-inflammatory activities that could promote inflammatory regression through inhibiting activation of LPS-induced macrophages [40]. In our previous study on Qiandongnan Xiaoxiang chickens, inflammation-related metabolites, L-Tryptophan and indole-3-carboxylic acid, were identified in low-weight chickens, and the inflammatory factor IL-6 that could inhibit the growth of chickens was found [20]. Therefore, we speculate the possible presence of unobserved inflammation in low-body-weight chickens and, as a response to inflammation and as a consequence, anti-inflammatory-related FAHFA levels were increased. 

It is worth noting that, in addition to inflammation-related metabolites, aflatoxin g1 and the steroid hormone 17beta-nandrolone decanoate were also enriched in serum of the low-body-weight chickens. Aflatoxins are a class of secondary metabolites synthesized by *Aspergillus flavus* with strong toxicity and high carcinogenicity [41]. In aflatoxin-fed mice, lower levels of IL-2, TNF-α, IL-17, and IFN-γ were found in the spleen and sera along with a reduced response of lymphocyte subsets and cytokine production [42]. The aflatoxin source was most likely mildew of feed, which reduces the growth, production, and antioxidant properties of broilers [43] resulting in liver damage, malnutrition, and weakened immunity [44]. The testosterone ortholog 17beta-nandrolone decanoate is an anabolic steroid that slows cell growth, inhibits mitochondrial respiration and respiratory-chain complexes I and III, and increases production of mitochondrial reactive oxygen species [45]. We found higher levels of 17 beta-nandrolone decanoate in the WL group and this may have directly influenced the growth of chickens. Similarly, in two other studies, our team also found that some sex hormones, such as pregnanetriol and 11beta-hydroxyprogesterone, were enriched in the serum of low-weight chickens. This may suggest that, after sexual maturity, physiological functions of chickens shift from growth to reproduction, slowing down their growth rate [20,46]. Together, these are potential reasons for the failure of WL chickens to gain weight during the experimental period.

## 5. Conclusions

This study identified a total of 13 subclasses of metabolites in the serum of Wumeng black-bone chickens, and found that some metabolites, including riboflavin and 2-isopropylmalic acid, might be linked to increased growth performance of chickens. Adequate riboflavin levels might improve the growth and development of chickens by promoting pathways such as acetyl group transfer into mitochondria and the TCA cycle. Inflammation might be an important reason for inhibition of the growth performance of chickens, and metabolites such as prostaglandin a2, kynurenic acid and FAHFA as well as their participation in enrichment of arachidonic acid and tryptophan metabolism might play crucial roles. The results of this study provide a reference for the analysis of growth traits of chickens and outlined a new direction for future research on improvement of growth performance of low-body-weight chickens in production practice.

Since (1) the sample size was limited, (2) only one breed was involved, and (3) validation experiments were not carried out, there are some limitations in this study. Therefore, further studies need to be conducted with larger populations and more breeds to identify key metabolites that are generally applicable to different chicken breeds. Meanwhile, it is necessary to feed chickens with the key metabolites to confirm the role of these metabolites in the growth and development of chickens.

## Figures and Tables

**Figure 1 animals-14-02743-f001:**
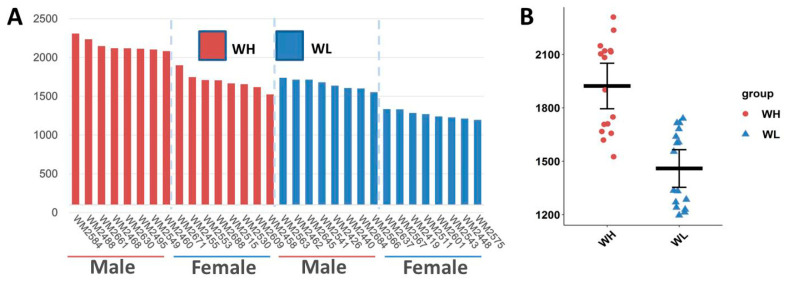
Body weights of chickens in the high- and low-body-weight groups. (**A**) Body weight of chickens at 160 days of age. (**B**) Comparison of body weight between the WH/WL groups. WH: high-body-weight group, *n* = 16; WL: low-body-weight group, *n* = 16.

**Figure 2 animals-14-02743-f002:**
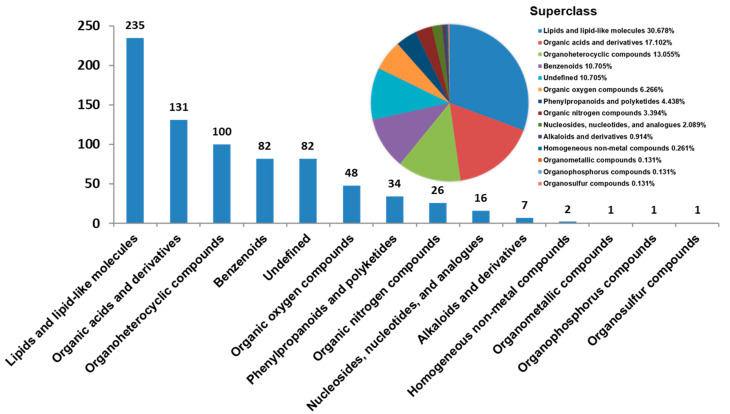
Metabolite types in the serum metabolome of Wumeng black-bone chickens. A total of 13 types of metabolites and some unclassified metabolites were identified. The number above the column is the number of metabolites of each type. The pie chart at the top right illustrates the proportion of metabolites in each superclass.

**Figure 3 animals-14-02743-f003:**
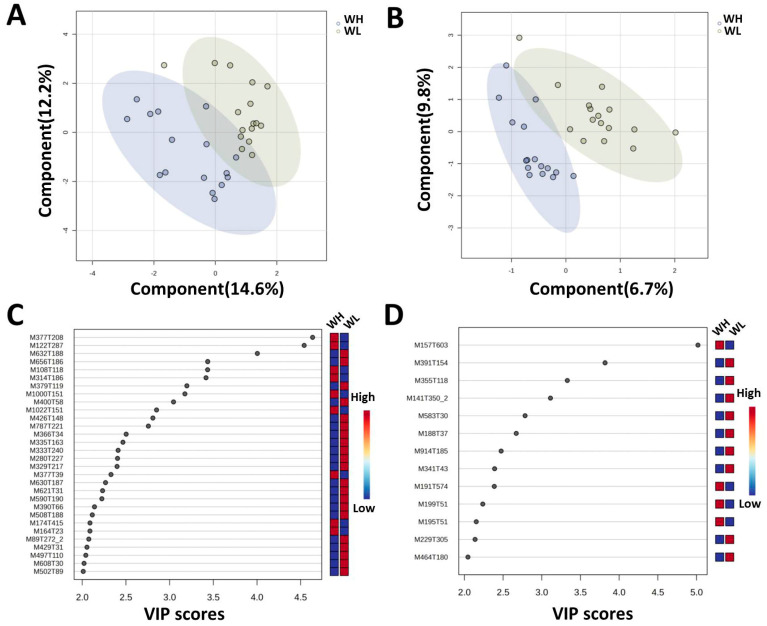
Analysis of serum metabolome of chickens in the WH and WL groups. PLS-DA analysis of non-targeted serum metabolites of chickens in the WH (*n* = 16) and WL (*n* = 16) group in (**A**) positive and (**B**) negative ion modes. Differential serum metabolites between the WH and WL groups and their VIP scores in (**C**) positive and (**D**) negative ion modes.

**Figure 4 animals-14-02743-f004:**
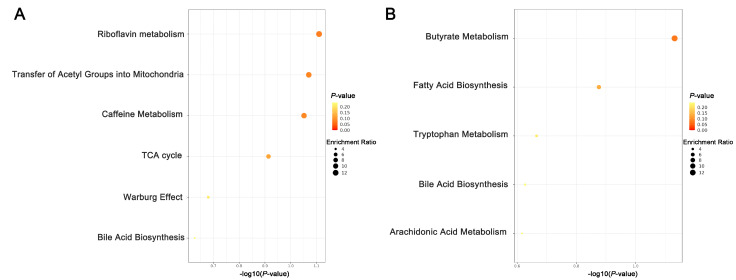
Enrichment analysis of metabolic pathways of serum differential metabolites in high- and low-body-weight groups. (**A**) High-body-weight group; (**B**) low-body-weight group.

**Table 1 animals-14-02743-t001:** Ingredient composition of the diets for chickens (air-dried basis).

Item	0–6 Weeks	6–18 Weeks
Ingredients, %		
Corn	56.30	58.62
Soybean meal	18.52	25.00
Rapeseed meal	10.00	0.00
Corn gluten meal	6.33	3.05
Wheat bran	2.94	5.63
Soybean oil	1.63	3.34
Limestone	1.18	1.18
Phytase	0.04	0.00
Choline chloride	0.15	0.00
Methionine	0.15	0.10
Lysine	0.22	0.32
NaCl	0.15	0.15
CaHPO_4_	1.89	1.61
Premix ^1^	0.05	1.00
total	100.00	100.00
Nutrients ^2^, %		
CP	21.18	19.05
ME, MJ/kg	12.12	12.56
Ca	1.0	0.90
AP	0.45	0.40
Met + Cys	0.90	0.72
Lys	1.06	0.90

^1^ Provides per kg of diet: For 0–6 weeks: Vitamin A, 6000 IU; Vitamin B_1_, 2.0 mg; Vitamin B_2_, 4.0 mg; Vitamin B5, 42 mg; Vitamin B_6_, 4.0 mg; Vitamin B_12_, 0.01 mg; Vitamin D_3_, 2000 IU; Vitamin E 30 IU; Vitamin K3, 1.8 mg; calcium pantothenate, 10.0 mg; biotin, 0.15 mg; folic acid, 0.85 mg; Fe, 80 mg; Cu, 8.0 mg; Mn, 80 mg; Zn, 65 mg; I, 0.50 mg; Se, 0.25 mg. For 6–18 weeks: Vitamin A, 10,000 IU; Vitamin B_1_, 2.50 mg; Vitamin B_2_, 7.5 mg; Vitamin B_6_, 3 mg; Vitamin D, 2000 IU; Vitamin E, 25 mg; Vitamin K, 2.8 mg; nicotinamide 40 mg; calcium pantothenate, 25 mg; biotin 0.20 mg; folic acid 1.5 mg; Vitamin B_12_, 0.015 mg, Fe, 80 mg; Cu, 8 mg, Mn, 100 mg; Zn, 60 mg; I, 0.35 mg; Se 0.3 mg. ^2^ CP was a measured value, while the others were calculated values.

**Table 2 animals-14-02743-t002:** Metabolites enriched in serum of high-body-weight chickens in positive ion mode.

ID	Enriched Groups	Annotated Metabolites	VIP Value
M377T208	WH	(-)-riboflavin	4.6368
M122T287	WH	Benzamide	4.5404
M108T118	WH	3-pyridinecarboxaldehyde	3.435
M314T186	WH	1-methyl-2-undecylquinolin-4-one	3.4166
M1000T151	WH	Taurochenodeoxycholic acid	3.1751
M1022T151	WH	2-((4r)-4-((3r,5r,6s,9s,10r,13r,14s,17r)-3,6-dihydroxy-10,13-dimethylhexadecahydro-1Hcyclopenta[a]phenanthren-17-yl)pentanamido)ethane-1-sulfonic acid	2.8506
M377T39	WH	1H-indazole-3-carboxylic acid, 1-(5-fluoropentyl)-, 1-naphthalenyl ester	2.329
M174T415	WH	Allidochlor	2.0885
M164T23	WH	1-deoxynojirimycin	2.088
M632T188	WL	2-erahpa [dmed-fahfa]	4.0043
M656T186	WL	2-epahsa [dmed-fahfa]	3.4361
M379T119	WL	Pyridate	3.1975
M400T58	WL	Demissidine	3.0447
M426T148	WL	N-[(3s,5s,7s)-adamantan-1-yl]-1-(4-fluorobenzyl)-1h-indazole-3-carboxamide	2.8085
M787T221	WL	Digitoxin	2.7568
M366T34	WL	Myriocin	2.5026
M335T163	WL	Prostaglandin a2	2.4657
M333T240	WL	Butamifos	2.4105
M280T227	WL	Oxamniquine	2.4038
M329T217	WL	Aflatoxin g1	2.3985
M630T187	WL	2-arahpa [dmed-fahfa]	2.2664
M621T31	WL	Ginsenoside rh1	2.2327
M590T190	WL	3-alahpda [dmed-fahfa]	2.2244
M390T66	WL	N-octanoylsphingosine	2.1401
M508T188	WL	1-(1z-octadecenyl)-sn-glycero-3-phosphocholine	2.1151
M89T272_2	WL	Butanoic acid	2.0747
M429T31	WL	17beta-nandrolone decanoate	2.0539
M497T110	WL	Acetamide, 2-[4-[(5,6-diphenyl-2-pyrazinyl)(1-methylethyl)amino]butoxy]-n-(methylsulfonyl)-	2.0406
M608T30	WL	1-lignoceroyl-2-hydroxy-sn-glycero-3-phosphocholine	2.0207
M502T89	WL	N-palmitoylsphingosine	2.0114

**Table 3 animals-14-02743-t003:** Metabolites enriched in serum of high-body-weight chickens in negative ion mode.

ID	Enriched Groups	Annotated Metabolites	VIP Value
M157T603	WH	2isopropylmalic acid	5.0179
M191T574	WH	Citrate	2.3865
M199T51	WH	Acetic acid, 2-(4-chloro-2-methylphenoxy)-	2.2378
M195T51	WH	3,7-dimethyluric acid	2.1533
M391T154	WL	Ursodeoxycholic acid	3.8177
M355T118	WL	Pioglitazone	3.3302
M141T350_2	WL	Kojic acid	3.1123
M583T30	WL	Ser-Leu-Ile-Gly-Lys-Val-amide	2.7844
M188T37	WL	Kynurenic acid	2.6701
M914T185	WL	Pi 40:4	2.4751
M341T43	WL	17-keto-4(z),7(z),10(z),13(z),15(e),19(z)-docosahexaenoic acid	2.3905
M229T305	WL	7-demethylsuberosin	2.1381
M464T180	WL	Glycocholic acid	2.0461

## Data Availability

The data presented in this study are available on request from the corresponding author.

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
