# Peer review of "Non-Targeted Metabolomics of Serum Reveals Biomarkers Associated with Body Weight in Wumeng Black-Bone Chickens"

_animals, 2024, doi:10.3390/ani14182743_

Round 1
Reviewer 1 Report
Comments and Suggestions for Authors
The study provides novel insights into the relationship between serum metabolites and growth performance in Chinese Wumeng black bone chickens using non-targeted metabolomics. The authors identified 13 differential metabolites and their associated pathways that could potentially influence the growth and development of chickens. The findings suggest that inflammation might be a key factor inhibiting growth performance, while adequate riboflavin could promote it. However, there are a few limitations to this study.
Major concern:
1. The Wumeng black-boned chicken may be a breed that has not undergone long-term commercial selective breeding, which is reflected in the experimental data showing a wide range of body weight variations within the population. This raises the question of whether this variation also implies a complex genetic background. Considering the relatively small total sample size and the limited number of experimental individuals included in this study, the reliability of the data and conclusions drawn from this research may have certain limitations.
2. The study is restricted to a single breed of chickens (Wumeng black bone), and further validation across multiple breeds is necessary. To ensure the reliability of the study's conclusions, I suggest that the authors provide additional experimental evidence by identifying more conservative influencing factors among the existing metabolites. This approach will help to validate the findings and demonstrate their applicability across different chicken breeds.
3. The experimental data show significant body weight differences between female and male individuals in both the WH and WL groups. However, the authors did not analyze or discuss the data separately for female and male individuals in the paper. I am curious about why the data was not divided into four groups for subsequent analysis and discussion: WH-female, WH-male, WL-female, and WL-male. Would this approach lead to more valuable conclusions?
4. The authors identified riboflavin and inflammation-related metabolites as potential key factors affecting chicken growth and development through data mining. However, this is far from sufficient. The authors should provide more in-depth studies or validation experiments to support their findings. Moreover, some targeted validation experiments should be conducted to confirm the roles of these metabolites in chicken growth and development.
Minor concern:
1. The title is "Non-targeted metabolomics of serum reveals biomarkers associated with body weight in Chinese local chickens", but in fact, only one breed was used in the study. It is suggested to directly use the name of the breed to accurately describe the research situation.
2. Line 74-76, the sentence contains grammatical redundancies and could benefit from simplification and improved punctuation for better clarity and readability.
3. Line 103-110 & Line 133-139, The font size and line spacing are inconsistent with the rest of the manuscript. Please adjust them accordingly.
4. Line 143-144, the manuscript mentions that "The Shapiro-Wilk normal distribution test indicated that the body weights did not conform to a normal distribution." Please explain this situation.
5. Line 150-151, Figure legend 1 does not provide an explanation for the abbreviation "WL".
6. Line 216, In the section "Enrichment analysis of metabolic pathways of differential serum metabolites", it is recommended that the authors provide more detailed information through supplementary materials.
Comments on the Quality of English LanguageModerate editing of English language required.
Author Response
Comment 1: The Wumeng black-boned chicken may be a breed that has not undergone long-term commercial selective breeding, which is reflected in the experimental data showing a wide range of body weight variations within the population. This raises the question of whether this variation also implies a complex genetic background. Considering the relatively small total sample size and the limited number of experimental individuals included in this study, the reliability of the data and conclusions drawn from this research may have certain limitations.
Response 1: Thank you very much. Indeed, as you said, there are certain limitations in relatively small total sample size and the limited number of experimental individuals, so we have added the limitation statement in the conclusion section. And we will conduct further research in larger groups in the future following your suggestion.
Comment 2: The study is restricted to a single breed of chickens (Wumeng black bone), and further validation across multiple breeds is necessary. To ensure the reliability of the study's conclusions, I suggest that the authors provide additional experimental evidence by identifying more conservative influencing factors among the existing metabolites. This approach will help to validate the findings and demonstrate their applicability across different chicken breeds.
Response 2: Thank you very much. Indeed, as you said, it is very necessary to conduct further validation across multiple breeds. In our previous study on other breeds including Qiandongnan Xiaoxiang chicken(Wang, L.; Zhang, F.; Li, H.; Yang, S.; Chen, X.; Long, S.; Yang, S.; Yang, Y.; Wang, Z. Metabolic and inflammatory linkage of the chicken cecal microbiome to growth performance. Front. Microbiol. 2023, 14, 1060458.)and Guizhou Yellow chicken (Yang, S., Y. Yang, X. Long, H. Li, F. Zhang, and Z. Wang. Integrated analysis of the effects of cecal microbiota and serum metabolome on market weights of chinese native chickens. 2023, Animals (Basel), 13, 19.), no metabolites identical to those in this study have been found. This suggested that growth-related metabolites may not be consistent among different breeds, and also indicating the complexity of chicken growth traits. Nevertheless, in our three studies we found that some metabolites with the same mechanism of action were associated with growth traits. For example, study on Wumeng black-boned chicken (this study) and Qiandongnan Xiaoxiang chicken (Metabolic and inflammatory linkage of the chicken cecal microbiome to growth performance) showed that inflammation is an important factor associated with inhibiting the growth of chickens, for metabolites related to inflammation were found in both breeds although the enriched metabolites were somewhat different. Specifically, L-Tryptophan and Indole-3-carboxylic acid enriched in low body weight Qiandongnan Xiaoxiang chicken, and prostaglandin a2, kynurenic acid and FAHFA as well as metabolism pathway arachidonic acid and tryptophan metabolism enriched in low body weight Wumeng black-bone chicken were all related to inflammation. Meanwhile, sex hormone enriched in low body weight group were found in all three breeds, such as Pregnanetriol in Qiandongnan Xiaoxiang Chicken, 11 beta-hydroxyprogeol in Guizhou Yellow Chicken, and 17-β-nandrolone decanoate in Wumeng black-bone chicken. We have added these contents in the discussion part. In addition, we are currently planning to conduct further study on other breeds of local chickens to obtain common metabolites among different breeds, so that to provide beneficial metabolites that are applicable to improve growth performance of different breeds of local chickens.
Comment 3: The experimental data show significant body weight differences between female and male individuals in both the WH and WL groups. However, the authors did not analyze or discuss the data separately for female and male individuals in the paper. I am curious about why the data was not divided into four groups for subsequent analysis and discussion: WH-female, WH-male, WL-female, and WL-male. Would this approach lead to more valuable conclusions?
Response 3: Thank you very much, but please excuse me. We followed the principle of half male and half female in the experiment design in order to eliminate the influence of gender, because the ultimate goal of our experiment is to find characteristic metabolites related to growth, with a view to applying them in production practice to promote the growth of Wumeng black bone chickens in the future. Therefore, common metabolites between different genders may be universal. So we did not analyze the data separately for female and male individuals.
Comment 4: The authors identified riboflavin and inflammation-related metabolites as potential key factors affecting chicken growth and development through data mining. However, this is far from sufficient. The authors should provide more in-depth studies or validation experiments to support their findings. Moreover, some targeted validation experiments should be conducted to confirm the roles of these metabolites in chicken growth and development.
Response 4: Thank you very much. In our previous study in title of “Metabolic and inflammatory linkage of the chicken cecal microbiome to growth performance”, we found that chicken growth is closely related to inflammatory cytokines, which confirms the results of this study from another aspect. As for riboflavin, many previous studies have shown that it is related to growth. Following your constructive suggestion, we added these studies and discussed them in the revised manuscript, and we are conducting further research to validate these findings. The added part is as follows:
Riboflavin supplementation can improve energy metabolism efficiency in mice or humans, the lack of riboflavin in the diet inhibits while its addition to diets can significantly improve the growth performance of poultry [31]. A previous study had evaluated riboflavin in broiler diets under different nutritional conditions and concluded that the supplemental level of 3.6 mg/kg added in the initial 4 weeks of life was optimal [32].
In our previous study on Qiandongnan Xiaoxiang chickens, inflammation-related metabolites, L-Tryptophan and Indole-3-carboxylic acid, were identified in low-weight chickens, and inflammatory factor IL-6 was found that could inhibits the growth of chickens [20].
Comment 5: The title is "Non-targeted metabolomics of serum reveals biomarkers associated with body weight in Chinese local chickens", but in fact, only one breed was used in the study. It is suggested to directly use the name of the breed to accurately describe the research situation.
Response 5: Thank you very much. We have revised the title as follows:
Non-targeted metabolomics of serum reveals biomarkers associated with body weight in Wumeng black bone chickens
Comment 6: Line 74-76, the sentence contains grammatical redundancies and could benefit from simplification and improved punctuation for better clarity and readability.
Response 6: Thank you very much. We have revised it as follows:
Chickens hatched in the same batch were reared under the same condition. Specifically, the animals were housed in three-layer iron cages with a feeding density of 16 mixed-sex chickens per cage at 0-4 weeks, 8 mixed-sex chickens per cage at 4-10 weeks and 1 chicken per cage at 10-18 weeks.
Comment 7: Line 103-110 & Line 133-139, The font size and line spacing are inconsistent with the rest of the manuscript. Please adjust them accordingly.
Response 7: Thank you very much. We have revised them.
Comment 8: Line 143-144, the manuscript mentions that "The Shapiro-Wilk normal distribution test indicated that the body weights did not conform to a normal distribution." Please explain this situation.
Response 8: Thank you very much. We carefully reviewed some literatures and found that in Shapiro-Wilk normal distribution test, P-value > 0.05 indicated a normal distribution. Our results showed that P-value = 0.1, which is > 0.05, indicating that the body weights conformed to a normal distribution. We are sorry that we have made a mistake in the original manuscript. And thank you so much for your reminding. We have revised it as follows:
The Shapiro-Wilk normal distribution test indicated that the body weights conformed to a normal distribution.
Comment 9: Line 150-151, Figure legend 1 does not provide an explanation for the abbreviation "WL".
Response 9: Thank you very much. We have added the explanation for the abbreviation "WL" as follows:
WL: low body weight group, n=16.
Comment 10: Line 216, In the section "Enrichment analysis of metabolic pathways of differential serum metabolites", it is recommended that the authors provide more detailed information through supplementary materials.
Response 10: Thank you very much. We have added supplementary materials containing more detailed information.

Reviewer 2 Report
Comments and Suggestions for Authors
This study compared the differences of serum metabolites in Wumeng black bone chickens with high and low weight at 160 days of age, and some biomarker metabolites in serum associated with growth performance of chickens were identified. But it could be refined for better readability and grammatical accuracy. Specific modification suggestions are as follows:
1. Lines 12-13: Add the full name of "UHPLC-MS/MS". The first time an abbreviation appears in the abstract and any part other than the abstract, its full name must be stated.
2. Lines 17-18: Replace "such as lipids and lipid-like molecules, organic acids and derivatives and organoheterocyclic compounds" with "such as lipids and lipid-like molecules, organic acids and derivatives, and organoheterocyclic compounds". When three or more nouns appear side by side, a comma should be added before conjunctions such as "and", "or", and "as well as". There are many similar errors in the manuscript, revising all of them.
3. Lines 27-28: It would be helpful to replace "WGCNA and Enrichment analysis" with "riboflavin, 2-isopropylmalic acid, and Inflammation" to improve the retrievability of the article.
4. Line 31: Replace "will undoubtably serve" with "will be undoubtably served".
5. Line 35: Delete the "or genes".
6. Line 38: Replace "mediator complex gene MED4" with "mediator complex gene named MED4". In addition, MLNR encodes the motilin receptor, not motilin.
7. Lines 59-60: "However, previous studies have not been consistent across breeds". Where does this view come from?
8. Lines 74-75: Delete the "in the same batch" or "on the same day".
9. Line 80: Replace "15-35°C" with "15-35 °C". There are many similar errors in the manuscript, revising all of them.
10. Line 82: Delete the "twice".
11. In the Table 1, replace "ME,MJ/kg" with "ME, MJ/kg". There are many similar errors in the manuscript, especially in the comment information of Table 1, revising all of them.
12. Line 96: Replace "methanol / acetonitrile / H2O (2:2:1 v/v)" with " methanol/acetonitrile/H2O (v/v/v: 2:2:1)". There are many similar errors in the manuscript, revising all of them.
13. Line 101: Replace "chromatography- mass" with "chromatography-mass".
14. Lines 107-108: "0-0.5 min, 95% B; 0.5-7.0 min, 95%-65% B; 7.0-8.0 min, 65%-40% B; 8.0-9.0 min, 40% B; 9-9.1 min, B 40%-95%; 9.1-12 min, 95% B". Replace all the "B" in this sentence with "mobile phase B".
15. Line 120: Delete the ":".
16. Line 136: Except for the first occurrence, it would be helpful to replace "high and low groups" with "WH and WL groups".
17. In the Figure 1A, it would be helpful to supplement the sex information of individuals.
18. Line 150: Missing annotation information for "WL".
19. Lines 168-185: Why does the author show this partial result? This part of the result does not contribute to the overall content and conclusion of the paper, and is very redundant. In addition, the comment information for "A" and "B" in Figure 3 is reversed.
20. Line 192: Replace "groups WH and WL" with "WH and WL groups".
21. Line 198: Replace "LW" with "WL". There are many similar errors in the manuscript, revising all of them.
22. Lines 216-226: How did the authors carry out the enrichment analysis? Why are the pathways in WH and WL groups so different? In addition, from the description in lines 217-223, the author said that carried out functional enrichment analysis of metabolites in WH and WL groups, respectively, while in line 225, it is pointed out that the functional enrichment analysis of the different metabolites between WH and WL groups is carried out. Which statement is correct?
23. Lines 231-232: "In contrast, compared with highly bred commercial breeds, are slow growers and this restricts industrial production and large-scale promotion of local chickens." Provide references to this view.
24. Line 240: What exactly are the 14 metabolites to which the author refers? Where do they come from? Why are there only 13 metabolites on line 249?
25. Line 265: Delete the "for".
26. Line 271: Delete the "a".
28. Line 276: Delete the "," in the "2-(4-chloro-2-methylphenoxy)-acetic acid,".
29. Line 62: Replace "black-bone" with "black bone".
Author Response
Comment 1: Lines 12-13: Add the full name of "UHPLC-MS/MS". The first time an abbreviation appears in the abstract and any part other than the abstract, its full name must be stated.
Response 1: Thank you very much. We have added the full name of "UHPLC-MS/MS" as follows:
In this study, we utilized non-targeted metabolomics using ultra-high performance liquid phase tandem mass spectrometry (UHPLC-MS/MS).
Comment 2: Lines 17-18: Replace "such as lipids and lipid-like molecules, organic acids and derivatives and organoheterocyclic compounds" with "such as lipids and lipid-like molecules, organic acids and derivatives, and organoheterocyclic compounds". When three or more nouns appear side by side, a comma should be added before conjunctions such as "and", "or", and "as well as". There are many similar errors in the manuscript, revising all of them.
Response 2: Thank you very much. We have revised them all through the manuscript following your suggestion.
Comment 3: Lines 27-28: It would be helpful to replace "WGCNA and Enrichment analysis" with "riboflavin, 2-isopropylmalic acid, and Inflammation" to improve the retrievability of the article.
Response 3: Thank you very much. We have replaced them.
Comment 4: Line 31: Replace "will undoubtably serve" with "will be undoubtably served".
Response 4: Thank you very much. We have replaced it.
Comment 5: Line 35: Delete the "or genes".
Response 5: Thank you very much. We have deleted it.
Comment 6: Line 38: Replace "mediator complex gene MED4" with "mediator complex gene named MED4". In addition, MLNR encodes the motilin receptor, not motilin.
Response 6: Thank you very much. We have revised them.
Comment 7: Lines 59-60: "However, previous studies have not been consistent across breeds". Where does this view come from?
Response 7: Thank you very much. This view comes from the cited literature [22] [20] [23]. We have revised the description as follows to make it clearer:
These studies have not been consistent so that metabolic mechanisms associated with body weight have not been fully revealed.
Comment 8: Lines 74-75: Delete the "in the same batch" or "on the same day".
Response 8: Thank you very much. We have deleted it.
Comment 9: Line 80: Replace "15-35°C" with "15-35 °C". There are many similar errors in the manuscript, revising all of them.
Response 9: Thank you very much. We have revised them all through the manuscript.
Comment 10: Line 82: Delete the "twice".
Response 10: Thank you very much. We have deleted it.
Comment 11: There are many similar errors in the manuscript, especially in the comment information of Table 1, revising all of them.
Response 11: Thank you very much. We have revised all of them.
Comment 12: Line 96: Replace "methanol / acetonitrile / H2O (2:2:1 v/v)" with " methanol/acetonitrile/H2O (v/v/v: 2:2:1)". There are many similar errors in the manuscript, revising all of them.
Response 12: Thank you very much. We have revised them all through the manuscript.
Comment 13: Line 101: Replace "chromatography- mass" with "chromatography-mass".
Response 13: Thank you very much. We have replaced it.
Comment 14: Lines 107-108: "0-0.5 min, 95% B; 0.5-7.0 min, 95%-65% B; 7.0-8.0 min, 65%-40% B; 8.0-9.0 min, 40% B; 9-9.1 min, B 40%-95%; 9.1-12 min, 95% B". Replace all the "B" in this sentence with "mobile phase B".
Response 14: Thank you very much. We have replaced all of them.
Comment 15: Line 120: Delete the ":".
Response 15: Thank you very much. We have deleted it.
Comment 16: Line 136: Except for the first occurrence, it would be helpful to replace "high and low groups" with "WH and WL groups".
Response 16: Thank you very much. We have replaced it.
Comment 17: In the Figure 1A, it would be helpful to supplement the sex information of individuals.
Response 17: Thank you very much. We have supplemented the sex information of individuals in the Figure 1A.
Comment 18: Line 150: Missing annotation information for "WL".
Response 18: Thank you very much. We have added the annotation information for "WL".
Comment 19: Lines 168-185: Why does the author show this partial result? This part of the result does not contribute to the overall content and conclusion of the paper, and is very redundant. In addition, the comment information for "A" and "B" in Figure 3 is reversed.
Response 19: Thank you very much. We have deleted this part.
Comment 20: Line 192: Replace "groups WH and WL" with "WH and WL groups".
Response 20: Thank you very much. We have replaced it.
Comment 21: Line 198: Replace "LW" with "WL". There are many similar errors in the manuscript, revising all of them.
Response 21: Thank you very much. We have replaced them all through the manuscript.
Comment 22: Lines 216-226: How did the authors carry out the enrichment analysis? Why are the pathways in WH and WL groups so different? In addition, from the description in lines 217-223, the author said that carried out functional enrichment analysis of metabolites in WH and WL groups, respectively, while in line 225, it is pointed out that the functional enrichment analysis of the different metabolites between WH and WL groups is carried out. Which statement is correct?
Response 22: Thank you very much. In order to explore the pathways in which the differential metabolites between WH and WL groups are enriched and thus affect body-weight, we performed Metabolite Set Enrichment Analysis (MSEA) on the differential metabolites in the WH and WL group, respectively. Specifically, KEGG IDs of the differential metabolites in each group were separately placed into the "Enrichment analysis" in MetaboAnalyst 6.0 to obtain the enrichment pathways. Therefore, the pathways in WH and WL groups were so different.
In addition, the description in lines 217-223 is unclear, and the description in line 225 is correct. So we revised the description in lines 217-223 as follows:
The functional enrichment analysis of the differential metabolites between WH and WL groups indicated that 6 metabolic pathways were enriched in serum of chickens in the WH group including riboflavin metabolism, transfer of acetyl groups into mitochondria, caffeine metabolism, citric acid (TCA) cycle, Warburg effect, and bile acid biosynthesis, while there were 5 metabolic pathways enriched for the WL group including inflammatory metabolic pathways (tryptophan and arachidonic acid metabolism) as well as butyrate metabolism and fatty acid and bile acid biosynthesis (Figure 4 and Supplementary Table S2).
Comment 23: Lines 231-232: "In contrast, compared with highly bred commercial breeds, are slow growers and this restricts industrial production and large-scale promotion of local chickens." Provide references to this view.
Response 23: Thank you very much. We have provided the reference to this view as follows:
Productive Performances of Slow-Growing Chicken Breeds and Their Crosses with a Commercial Strain in Conventional and Free-Range Farming Systems
Comment 24: Line 240: What exactly are the 14 metabolites to which the author refers? Where do they come from? Why are there only 13 metabolites on line 249?
Response 24: Thank you very much for your reminding. We have checked the manuscript carefully and found that “13 metabolites” is correct. We have revised it.
Comment 25: Line 265: Delete the "for".
Response 25: Thank you very much. We have deleted it.
Comment 26: Line 271: Delete the "a".
Response 26: Thank you very much. We have deleted it.
Comment 27: Line 276: Delete the "," in the "2-(4-chloro-2-methylphenoxy)-acetic acid,".
Response 27: Thank you very much. We have deleted it.
Comment 28: Line 62: Replace "black-bone" with "black bone".
Response 28: Thank you very much. We have replaced it.

Round 2
Reviewer 2 Report
Comments and Suggestions for Authors
The author has carefully revised all my concerns, and the manuscript can be published in current form.
Author Response
Thank you very much for your comments.